# Decoupled Deep Neural Network for Semi-supervised Semantic Segmentation

**Seunghoon Hong**[*]   **Hyeonwoo Noh**[*]   **Bohyung Han**
Dept. of Computer Science and Engineering, POSTECH, Pohang, Korea
{maga33,hyeonwoonoh_,bhhan}@postech.ac.kr

## Abstract

We propose a novel deep neural network architecture for semi-supervised semantic segmentation using heterogeneous annotations. Contrary to existing approaches posing semantic segmentation as a single task of region-based classification, our algorithm decouples classification and segmentation, and learns a separate network for each task. In this architecture, labels associated with an image are identified by classification network, and binary segmentation is subsequently performed for each identified label in segmentation network. The decoupled architecture enables us to learn classification and segmentation networks separately based on the training data with image-level and pixel-wise class labels, respectively. It facilitates to reduce search space for segmentation effectively by exploiting class-specific activation maps obtained from bridging layers. Our algorithm shows outstanding performance compared to other semi-supervised approaches with much less training images with strong annotations in PASCAL VOC dataset.

## 1   Introduction

Semantic segmentation is a technique to assign structured semantic labels—typically, object class labels—to individual pixels in images. This problem has been studied extensively over decades, yet remains challenging since object appearances involve significant variations that are potentially originated from pose variations, scale changes, occlusion, background clutter, etc. However, in spite of such challenges, the techniques based on Deep Neural Network (DNN) demonstrate impressive performance in the standard benchmark datasets such as PASCAL VOC [1].

Most DNN-based approaches pose semantic segmentation as pixel-wise classification problem [2, 3, 4, 5, 6]. Although these approaches have achieved good performance compared to previous methods, training DNN requires a large number of segmentation ground-truths, which result from tremendous annotation efforts and costs. For this reason, reliable pixel-wise segmentation annotations are typically available only for a small number of classes and images, which makes supervised DNNs difficult to be applied to semantic segmentation tasks involving various kinds of objects.

Semi- or weakly-supervised learning approaches [7, 8, 9, 10] alleviate the problem in lack of training data by exploiting weak label annotations per bounding box [10, 8] or image [7, 8, 9]. They often assume that a large set of weak annotations is available during training while training examples with strong annotations are missing or limited. This is a reasonable assumption because weak annotations such as class labels for bounding boxes and images require only a fraction of efforts compared to strong annotations, *i.e.,* pixel-wise segmentations. The standard approach in this setting is to update the model of a supervised DNN by iteratively inferring and refining hypothetical segmentation labels using weakly annotated images. Such iterative techniques often work well in practice [8, 10], but training methods rely on ad-hoc procedures and there is no guarantee of convergence; implementation may be tricky and the algorithm may not be straightforward to reproduce.

---

[*]Both authors have equal contribution on this paper.

We propose a novel decoupled architecture of DNN appropriate for semi-supervised semantic segmentation, which exploits heterogeneous annotations with a small number of strong annotations—full segmentation masks—as well as a large number of weak annotations—object class labels per image. Our algorithm stands out from the traditional DNN-based techniques because the architecture is composed of two separate networks; one is for classification and the other is for segmentation. In the proposed network, object labels associated with an input image are identified by classification network while figure-ground segmentation of each identified label is subsequently obtained by segmentation network. Additionally, there are bridging layers, which deliver class-specific information from classification to segmentation network and enable segmentation network to focus on the single label identified by classification network at a time.

Training is performed on each network separately, where networks for classification and segmentation are trained with image-level and pixel-wise annotations, respectively; training does not require iterative procedure, and algorithm is easy to reproduce. More importantly, decoupling classification and segmentation reduces search space for segmentation significantly, which makes it feasible to train the segmentation network with a handful number of segmentation annotations. Inference in our network is also simple and does not involve any post-processing. Extensive experiments show that our network substantially outperforms existing semi-supervised techniques based on DNNs even with much smaller segmentation annotations, *e.g.,* 5 or 10 strong annotations per class.

The rest of the paper is organized as follows. We briefly review related work and introduce overall algorithm in Section 2 and 3, respectively. The detailed configuration of the proposed network is described in Section 4, and training algorithm is presented in Section 5. Section 6 presents experimental results on a challenging benchmark dataset.

## 2 Related Work

Recent breakthrough in semantic segmentation are mainly driven by supervised approaches relying on Convolutional Neural Network (CNN) [2, 3, 4, 5, 6]. Based on CNNs developed for image classification, they train networks to assign semantic labels to local regions within images such as pixels [2, 3, 4] or superpixels [5, 6]. Notably, Long *et al.* [2] propose an end-to-end system for semantic segmentation by transforming a standard CNN for classification into a fully convolutional network. Later approaches improve segmentation accuracy through post-processing based on fully-connected CRF [3, 11]. Another branch of semantic segmentation is to learn a multi-layer deconvolution network, which also provides a complete end-to-end pipeline [12]. However, training these networks requires a large number of segmentation ground-truths, but the collection of such dataset is a difficult task due to excessive annotation efforts.

To mitigate heavy requirement of training data, weakly-supervised learning approaches start to draw attention recently. In weakly-supervised setting, the models for semantic segmentation have been trained with only image-level labels [7, 8, 9] or bounding box class labels [10]. Given weakly annotated training images, they infer latent segmentation masks based on Multiple Instance Learning (MIL) [7, 9] or Expectation-Maximization (EM) [8] framework based on the CNNs for supervised semantic segmentation. However, performance of weakly supervised learning approaches except [10] is substantially lower than supervised methods, mainly because there is no direct supervision for segmentation during training. Note that [10] requires bounding box annotations as weak supervision, which are already pretty strong and significantly more expensive to acquire than image-level labels.

Semi-supervised learning is an alternative to bridge the gap between fully- and weakly-supervised learning approaches. In the standard semi-supervised learning framework, given only a small number of training images with strong annotations, one needs to infer the full segmentation labels for the rest of the data. However, it is not plausible to learn a huge number of parameters in deep networks reliably in this scenario. Instead, [8, 10] train the models based on heterogeneous annotations—a large number of weak annotations as well as a small number strong annotations. This approach is motivated from the facts that the weak annotations, *i.e.,* object labels per bounding box or image, is much more easily accessible than the strong ones and that the availability of the weak annotations is useful to learn a deep network by mining additional training examples with full segmentation masks. Based on supervised CNN architectures, they iteratively infer and refine pixel-wise segmentation labels of weakly annotated images with guidance of strongly annotated images, where image-level labels [8] and bounding box annotations [10] are employed as weak annotations. They

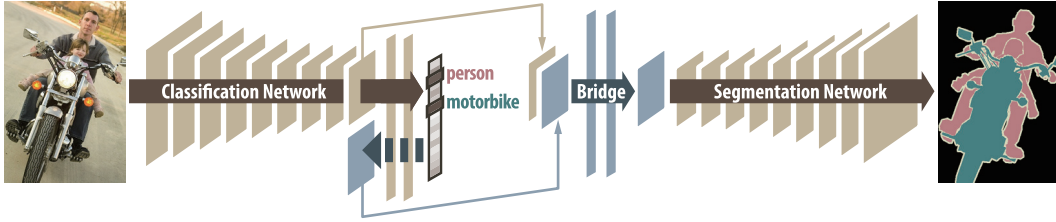

Figure 1: The architecture of the proposed network. While classification and segmentation networks are decoupled, bridging layers deliver critical information from classification network to segmentation network.

claim that exploiting few strong annotations substantially improves the accuracy of semantic segmentation while it reduces annotations efforts for supervision significantly. However, they rely on iterative training procedures, which are often ad-hoc and heuristic and increase complexity to reproduce results in general. Also, these approaches still need a fairly large number of strong annotations to achieve reliable performance.

## 3  Algorithm Overview

Figure 1 presents the overall architecture of the proposed network. Our network is composed of three parts: classification network, segmentation network and bridging layers connecting the two networks. In this model, semantic segmentation is performed by separate but successive operations of classification and segmentation. Given an input image, classification network identifies labels associated with the image, and segmentation network produces pixel-wise figure-ground segmentation corresponding to each identified label. This formulation may suffer from loose connection between classification and segmentation, but we figure out this challenge by adding bridging layers between the two networks and delivering class-specific information from classification network to segmentation network. Then, it is possible to optimize the two networks using separate objective functions while the two decoupled tasks collaborate effectively to accomplish the final goal.

Training our network is very straightforward. We assume that a large number of image-level annotations are available while there are only a few training images with segmentation annotations. Given these heterogeneous and unbalanced training data, we first learn the classification network using rich image-level annotations. Then, with the classification network fixed, we jointly optimize the bridging layers and the segmentation network using a small number of training examples with strong annotations. There are only a small number of strongly annotated training data, but we alleviate this challenging situation by generating many artificial training examples through data augmentation.

The contributions and characteristics of the proposed algorithm are summarized below:

- We propose a novel DNN architecture for semi-supervised semantic segmentation using heterogeneous annotations. The new architecture decouples classification and segmentation tasks, which enables us to employ pre-trained models for classification network and train only segmentation network and bridging layers using a few strongly annotated data.

- The bridging layers construct class-specific activation maps, which are delivered from classification network to segmentation network. These maps provide strong priors for segmentation, and reduce search space dramatically for training and inference.

- Overall training procedure is very simple since two networks are to be trained separately. Our algorithm does not infer segmentation labels of weakly annotated images through iterative heuristics[1], which are common in semi-supervised learning techniques [8, 10].

The proposed algorithm provides a concept to make up for the lack of strongly annotated training data using a large number of weakly annotated data. This concept is interesting because the assump-

_______________

[1]Due to this property, our framework is different from standard semi-supervised learning but close to few-shot learning with heterogeneous annotations. Nonetheless, we refer to it as *semi-supervised* learning in this paper since we have a fraction of strongly annotated data in our training dataset but complete annotations of weak labels. Note that our level of supervision is similar to the semi-supervised learning case in [8].

tion about the availability of training data is desirable for real situations. We estimate figure-ground segmentation maps only for the relevant classes identified by classification network, which improves scalability of algorithm in terms of the number of classes. Finally, our algorithm outperforms the comparable semi-supervised learning method with substantial margins in various settings.

## 4   Architecture

This section describes the detailed configurations of the proposed network, including classification network, segmentation network and bridging layers between the two networks.

### 4.1   Classification Network

The classification network takes an image $\mathbf{x}$ as its input, and outputs a normalized score vector $S(\mathbf{x}; \theta_c) \in \mathrm{R}^L$ representing a set of relevance scores of the input $\mathbf{x}$ based on the trained classification model $\theta_c$ for predefined $L$ categories. The objective of classification network is to minimize error between ground-truths and estimated class labels, and is formally written as

$$\min_{\theta_c} \sum_i e_c(\mathbf{y}_i, S(\mathbf{x}_i; \theta_c)), \tag{1}$$

where $\mathbf{y}_i \in \{0, 1\}^L$ denotes the ground-truth label vector of the $i$-th example and $e_c(\mathbf{y}_i, S(\mathbf{x}_i; \theta_c))$ is classification loss of $S(\mathbf{x}_i; \theta_c)$ with respect to $\mathbf{y}_i$.

We employ VGG 16-layer net [13] as the base architecture for our classification network. It consists of 13 convolutional layers, followed by rectification and optional pooling layers, and 3 fully connected layers for domain-specific projection. Sigmoid cross-entropy loss function is employed in Eq. (1), which is a typical choice in multi-class classification tasks.

Given output scores $S(\mathbf{x}_i; \theta_c)$, our classification network identifies a set of labels $\mathcal{L}_i$ associated with input image $\mathbf{x}_i$. The region in $\mathbf{x}_i$ corresponding to each label $l \in \mathcal{L}_i$ is predicted by the segmentation network discussed next.

### 4.2   Segmentation Network

The segmentation network takes a class-specific activation map $\mathbf{g}_i^l$ of input image $\mathbf{x}_i$, which is obtained from bridging layers, and produces a two-channel class-specific segmentation map $M(\mathbf{g}_i^l; \theta_s)$ after applying softmax function, where $\theta_s$ is the model parameter of segmentation network. Note that $M(\mathbf{g}_i^l; \theta_s)$ has foreground and background channels, which are denoted by $M_f(\mathbf{g}_i^l; \theta_s)$ and $M_b(\mathbf{g}_i^l; \theta_s)$, respectively. The segmentation task is formulated as per-pixel regression to ground-truth segmentation, which minimizes

$$\min_{\theta_s} \sum_i e_s(\mathbf{z}_i^l, M(\mathbf{g}_i^l; \theta_s)), \tag{2}$$

where $\mathbf{z}_i^l$ denotes the binary ground-truth segmentation mask for category $l$ of the $i$-th image $\mathbf{x}_i$ and $e_s(\mathbf{z}_i, M(\mathbf{g}_i^l; \theta_s))$ is the segmentation loss of $M_f(\mathbf{g}_i^l; \theta_s)$—or equivalently the segmentation loss of $M_b(\mathbf{g}_i^l; \theta_s)$—with respect to $\mathbf{z}_i^l$.

The recently proposed deconvolution network [12] is adopted for our segmentation network. Given an input activation map $\mathbf{g}_i^l$ corresponding to input image $\mathbf{x}_i$, the segmentation network generates a segmentation mask in the same size to $\mathbf{x}_i$ by multiple series of operations of unpooling, deconvolution and rectification. Unpooling is implemented by importing the switch variable from every pooling layer in the classification network, and the number of deconvolutional and unpooling layers are identical to the number of convolutional and pooling layers in the classification network. We employ the softmax loss function to measure per-pixel loss in Eq. (2).

Note that the objective function in Eq. (2) corresponds to pixel-wise *binary* classification; it infers whether each pixel belongs to the given class $l$ or not. This is the major difference from the existing networks for semantic segmentation including [12], which aim to classify each pixel to one of the $L$ predefined classes. By decoupling classification from segmentation and posing the objective of segmentation network as binary classification, our algorithm reduces the number of parameters

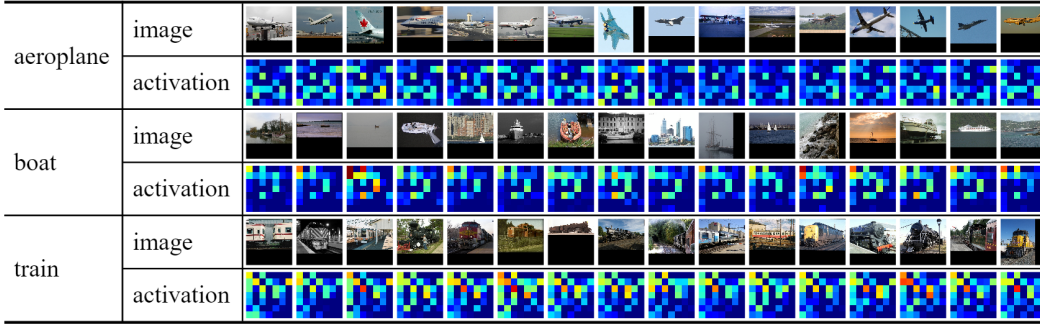

Figure 2: Examples of class-specific activation maps (output of bridging layers). We show the most representative channel for visualization. Despite significant variations in input images, the class-specific activation maps share similar properties.

in the segmentation network significantly. Specifically, this is because we identify the relevant labels using classification network and perform binary segmentation for each of the labels, where the number of output channels in segmentation network is set to two—for foreground and background—regardless of the total number of candidate classes. This property is especially advantageous in our challenging scenario, where only a few pixel-wise annotations (typically 5 to 10 annotations per class) are available for training segmentation network.

### 4.3 Bridging Layers

To enable the segmentation network described in Section 4.2 to produce the segmentation mask of a specific class, the input to the segmentation network should involve class-specific information as well as spatial information required for shape generation. To this end, we have additional layers underneath segmentation network, which is referred to as bridging layers, to construct the class-specific activation map $\mathbf{g}_i^l$ for each identified label $l \in \mathcal{L}_i$.

To encode spatial configuration of objects presented in image, we exploit outputs from an intermediate layer in the classification network. We take the outputs from the last pooling layer (pool5) since the activation patterns of convolution and pooling layers often preserve spatial information effectively while the activations in the higher layers tend to capture more abstract and global information. We denote the activation map of pool5 layer by $\mathbf{f}_{\text{spat}}$ afterwards.

Although activations in $\mathbf{f}_{\text{spat}}$ maintain useful information for shape generation, they contain mixed information of all relevant labels in $\mathbf{x}_i$ and we should identify class-specific activations in $\mathbf{f}_{\text{spat}}$ additionally. For the purpose, we compute class-specific saliency maps using the back-propagation technique proposed in [14]. Let $\mathbf{f}^{(i)}$ be the output of the $i$-th layer ($i = 1, \ldots, M$) in the classification network. The relevance of activations in $\mathbf{f}^{(k)}$ with respect to a specific class $l$ is computed by chain rule of partial derivative, which is similar to error back-propagation in optimization, as

$$\mathbf{f}_{\text{cls}}^l = \frac{\partial S_l}{\partial \mathbf{f}^{(k)}} = \frac{\partial \mathbf{f}^{(M)}}{\partial \mathbf{f}^{(M-1)}} \frac{\partial \mathbf{f}^{(M-1)}}{\partial \mathbf{f}^{(M-2)}} \cdots \frac{\partial \mathbf{f}^{(k+1)}}{\partial \mathbf{f}^{(k)}}, \tag{3}$$

where $\mathbf{f}_{\text{cls}}^l$ denotes class-specific saliency map and $S_l$ is the classification score of class $l$. Intuitively, Eq. (3) means that the values in $\mathbf{f}_{\text{cls}}^l$ depend on how much the activations in $\mathbf{f}^{(k)}$ are relevant to class $l$; this is measured by computing the partial derivative of class score $S_l$ with respect to the activations in $\mathbf{f}^{(k)}$. We back-propagate the class-specific information until pool5 layer.

The class-specific activation map $\mathbf{g}_i^l$ is obtained by combining both $\mathbf{f}_{\text{spat}}$ and $\mathbf{f}_{\text{cls}}^l$. We first concatenate $\mathbf{f}_{\text{spat}}$ and $\mathbf{f}_{\text{cls}}^l$ in their channel direction, and forward-propagate it through the fully-connected bridging layers, which discover the optimal combination of $\mathbf{f}_{\text{spat}}$ and $\mathbf{f}_{\text{cls}}^l$ using the trained weights. The resultant class-specific activation map $\mathbf{g}_i^l$ that contains both spatial and class-specific information is given to segmentation network to produce a class-specific segmentation map. Note that the changes in $\mathbf{g}_i^l$ depend only on $\mathbf{f}_{\text{cls}}^l$ since $\mathbf{f}_{\text{spat}}$ is fixed for all classes in an input image.

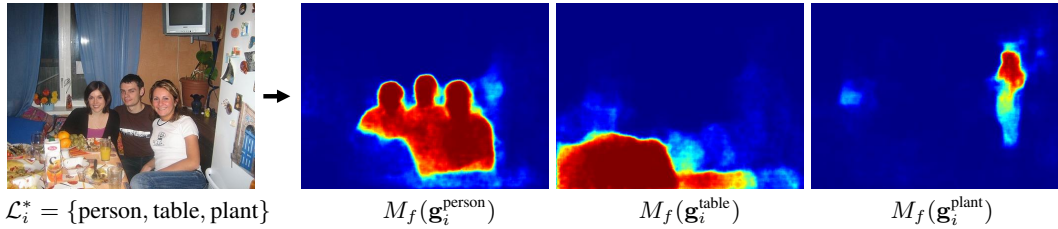

$\mathcal{L}_i^* = \{\text{person, table, plant}\}$      $M_f(\mathbf{g}_i^{\text{person}})$      $M_f(\mathbf{g}_i^{\text{table}})$      $M_f(\mathbf{g}_i^{\text{plant}})$

Figure 3: Input image (left) and its segmentation maps (right) of individual classes.

Figure 2 visualizes the examples of class-specific activation maps $\mathbf{g}_i^l$ obtained from several validation images. The activations from the images in the same class share similar patterns despite substantial appearance variations, which shows that the outputs of bridging layers capture class-specific information effectively; this property makes it possible to obtain figure-ground segmentation maps for individual relevant classes in segmentation network. More importantly, it reduces the variations of input distributions for segmentation network, which allows to achieve good generalization performance in segmentation even with a small number of training examples.

For inference, we compute a class-specific activation map $\mathbf{g}_i^l$ for each identified label $l \in \mathcal{L}_i$ and obtain class-specific segmentation maps $\{M(\mathbf{g}_i^l; \theta_s)\}_{\forall l \in \mathcal{L}_i}$. In addition, we obtain $M(\mathbf{g}_i^*; \theta_s)$, where $\mathbf{g}_i^*$ is the activation map from the bridging layers for all identified labels. The final label estimation is given by identifying the label with the maximum score in each pixel out of $\{M_f(\mathbf{g}_i^l; \theta_s)\}_{\forall l \in \mathcal{L}_i}$ and $M_b(\mathbf{g}_i^*; \theta_s)$. Figure 3 illustrates the output segmentation map of each $\mathbf{g}_i^l$ for $\mathbf{x}_i$, where each map identifies high response area given $\mathbf{g}_i^l$ successfully.

## 5 Training

In our semi-supervised learning scenario, we have mixed training examples with weak and strong annotations. Let $\mathcal{W} = \{1, ..., N_w\}$ and $\mathcal{S} = \{1, ..., N_s\}$ denote the index sets of images with image-level and pixel-wise class labels, respectively, where $N_w \gg N_s$. We first train the classification network using the images in $\mathcal{W}$ by optimizing the loss function in Eq. (1). Then, fixing the weights in the classification network, we jointly train the bridging layers and the segmentation network using images in $\mathcal{S}$ by optimizing Eq. (2). For training segmentation network, we need to obtain class-specific activation map $\mathbf{g}_i^l$ from bridging layers using ground-truth class labels associated with $\mathbf{x}_i$, $i \in \mathcal{S}$. Note that we can reduce complexity in training by optimizing the two networks separately.

Although the proposed algorithm has several advantages in training segmentation network with few training images, it would still be better to have more training examples with strong annotations. Hence, we propose an effective data augmentation strategy, *combinatorial cropping*. Let $\mathcal{L}_i^*$ denotes a set of ground-truth labels associated with image $\mathbf{x}_i, i \in \mathcal{S}$. We enumerate all possible combinations of labels in $\mathsf{P}(\mathcal{L}_i^*)$, where $\mathsf{P}(\mathcal{L}_i^*)$ denotes the powerset of $\mathcal{L}_i^*$. For each $\mathcal{P} \in \mathsf{P}(\mathcal{L}_i^*)$ except empty set ($\mathcal{P} \neq \emptyset$), we construct a binary ground-truth segmentation mask $\mathbf{z}_i^{\mathcal{P}}$ by setting the pixels corresponding to every label $l \in \mathcal{P}$ as foreground and the rests as background. Then, we generate $N_p$ sub-images enclosing the foreground areas based on region proposal method [15] and random sampling. Through this simple data augmentation technique, we have $N_t = N_s + N_p \cdot \left( \sum_{i \in \mathcal{S}} 2^{|\mathcal{L}_i^*|} - 1 \right)$ training examples with strong annotations effectively, where $N_t \gg N_s$.

## 6 Experiments

### 6.1 Implementation Details

**Dataset** We employ PASCAL VOC 2012 dataset [1] for training and testing of the proposed deep network. The dataset with extended annotations from [16], which contains 12,031 images with pixel-wise class labels, is used in our experiment. To simulate semi-supervised learning scenario, we divide the training images into two non-disjoint subsets—$\mathcal{W}$ with weak annotations only and $\mathcal{S}$ with strong annotations as well. There are 10,582 images with image-level class labels, which are used to train our classification network. We also construct training datasets with strongly annotated images;

Table 1: Evaluation results on PASCAL VOC 2012 validation set.

| # of strongs | DecoupledNet | WSSL-Small_FoV [8] | WSSL-Large-FoV [8] | DecoupledNet-Str | DeconvNet [12] |
|---|---|---|---|---|---|
| Full | 67.5 | 63.9 | **67.6** | 67.5 | 67.1 |
| 25 ($\times$20 classes) | **62.1** | 56.9 | 54.2 | 50.3 | 38.6 |
| 10 ($\times$20 classes) | **57.4** | 47.6 | 38.9 | 41.7 | 21.5 |
| 5  ($\times$20 classes) | **53.1** | - | - | 32.7 | 15.3 |

Table 2: Evaluation results on PASCAL VOC 2012 test set.

| Models | bkg | areo | bike | bird | boat | bottle | bus | car | cat | chair | cow | table | dog | horse | mbk | prsn | plnt | sheep | sofa | train | tv | mean |
|---|---|---|---|---|---|---|---|---|---|---|---|---|---|---|---|---|---|---|---|---|---|---|
| DecoupledNet-Full | 91.5 | 78.8 | 39.9 | 78.1 | 53.8 | 68.3 | 83.2 | 78.2 | 80.6 | 25.8 | 62.6 | 55.5 | 75.1 | 77.2 | 77.1 | 76.0 | 47.8 | 74.1 | 47.5 | 66.4 | 60.4 | 66.6 |
| DecoupledNet-25 | 90.1 | 75.8 | 41.7 | 70.4 | 46.4 | 66.2 | 83.0 | 69.9 | 76.7 | 23.1 | 61.2 | 43.3 | 70.4 | 75.7 | 74.1 | 65.7 | 46.2 | 73.8 | 39.7 | 61.9 | 57.6 | 62.5 |
| DecoupledNet-10 | 88.5 | 73.8 | 40.1 | 68.1 | 45.5 | 59.5 | 76.4 | 62.7 | 71.4 | 17.7 | 60.4 | 39.9 | 64.5 | 73.0 | 68.5 | 56.0 | 43.4 | 70.8 | 37.8 | 60.3 | 54.2 | 58.7 |
| DecoupledNet-5 | 87.4 | 70.4 | 40.9 | 60.4 | 36.3 | 61.2 | 67.3 | 67.7 | 64.6 | 12.8 | 60.2 | 26.4 | 63.2 | 69.6 | 64.8 | 53.1 | 34.7 | 65.3 | 34.4 | 57.0 | 50.5 | 54.7 |

the number of images with segmentation labels per class is controlled to evaluate the impact of supervision level. In our experiment, three different cases—5, 10, or 25 training images with strong annotations per class—are tested to show the effectiveness of our semi-supervised framework. We evaluate the performance of the proposed algorithm on 1,449 validation images.

**Data Augmentation**  We employ different strategies to augment training examples in the two datasets with weak and strong annotations. For the images with weak annotations, simple data augmentation techniques such as random cropping and horizontal flipping are employed as suggested in [13]. We perform combinatorial cropping proposed in Section 5 for the images with strong annotations, where EdgeBox [15] is adopted to generate region proposals and the $N_p(=200)$ sub-images are generated for each label combination.

**Optimization**  The proposed network is implemented based on Caffe library [17]. The standard Stochastic Gradient Descent (SGD) with momentum is employed for optimization, where all parameters are identical to [12]. We initialize the weights of the classification network using VGG 16-layer net pre-trained on ILSVRC [18] dataset. When we train the deep network with full annotations, the network converges after approximately 5.5K and 17.5K SGD iterations with mini-batches of 64 examples in training classification and segmentation networks, respectively; training takes 3 days (0.5 day for classification network and 2.5 days for segmentation network) in a single Nvidia GTX Titan X GPU with 12G memory. Note that training segmentation network is much faster in our semi-supervised setting while there is no change in training time of classification network.

## 6.2   Results on PASCAL VOC Dataset

Our algorithm denoted by DecoupledNet is compared with two variations in WSSL [8], which is another algorithm based on semi-supervised learning with heterogeneous annotations. We also test the performance of DecoupledNet-Str[2] and DeconvNet [12], which only utilize examples with strong annotations, to analyze the benefit of image-level weak annotations. All learned models in our experiment are based only on the training set (not including the validation set) in PASCAL VOC 2012 dataset. All algorithms except WSSL [8] report the results without CRF. Segmentation accuracy is measured by Intersection over Union (IoU) between ground-truth and predicted segmentation, and the mean IoU over 20 semantic categories is employed for the final performance evaluation.

Table 1 summarizes quantitative results on PASCAL VOC 2012 validation set. Given the same amount of supervision, DecoupledNet presents substantially better performance even without any post-processing than WSSL [8], which is a directly comparable method. In particular, our algorithm has great advantage over WSSL when the number of strong annotations is extremely small. We believe that this is because DecoupledNet reduces search space for segmentation effectively by employing the bridging layers and the deep network can be trained with a smaller number of images with strong annotations consequently. Our results are even more meaningful since training procedure of DecoupledNet is very straightforward compared to WSSL and does not involve heuristic iterative procedures, which are common in semi-supervised learning methods.

When there are only a small number of strongly annotated training data, our algorithm obviously outperforms DecoupledNet-Str and DeconvNet [12] by exploiting the rich information of weakly an-

| Input image | Ground-truth | 5 examples | 10 examples | 25 examples | Full annotations |
|---|---|---|---|---|---|

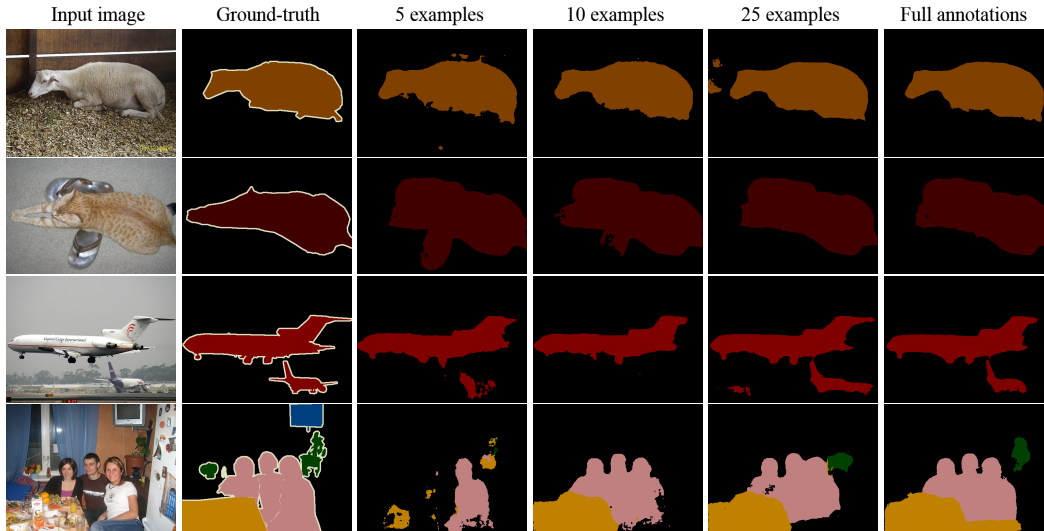

Figure 4: Semantic segmentation results of several PASCAL VOC 2012 validation images based on the models trained on a different number of pixel-wise segmentation annotations.

notated images. It is interesting that DecoupledNet-Str is clearly better than DeconvNet, especially when the number of training examples is small. For reference, the best accuracy of the algorithm based only on the examples with image-level labels is 42.0% [7], which is much lower than our result with five strongly annotated images per class, even though [7] requires significant efforts for heuristic post-processing. These results show that even little strong supervision can improve semantic segmentation performance dramatically.

Table 2 presents more comprehensive results of our algorithm in PASCAL VOC test set. Our algorithm works well in general and approaches to the empirical upper-bound fast with a small number of strongly annotated images. A drawback of our algorithm is that it does not achieve the state-of-the-art performance [3, 11, 12] when the (almost[3]) full supervision is provided in PASCAL VOC dataset. This is probably because our method optimizes classification and segmentation networks separately although joint optimization of two objectives is more desirable. However, note that our strategy is more appropriate for semi-supervised learning scenario as shown in our experiment.

Figure 4 presents several qualitative results from our algorithm. Note that our model trained only with five strong annotations per class already shows good generalization performance, and that more training examples with strong annotations improve segmentation accuracy and reduce label confusions substantially. Refer to our project website[4] for more comprehensive qualitative evaluation.

## 7 Conclusion

We proposed a novel deep neural network architecture for semi-supervised semantic segmentation with heterogeneous annotations, where classification and segmentation networks are decoupled for both training and inference. The decoupled network is conceptually appropriate for exploiting heterogeneous and unbalanced training data with image-level class labels and/or pixel-wise segmentation annotations, and simplifies training procedure dramatically by discarding complex iterative procedures for intermediate label inferences. Bridging layers play a critical role to reduce output space of segmentation, and facilitate to learn segmentation network using a handful number of segmentation annotations. Experimental results validate the effectiveness of our decoupled network, which outperforms existing semi- and weakly-supervised approaches with substantial margins.

**Acknowledgement** This work was partly supported by the ICT R&D program of MSIP/IITP [B0101-15-0307; ML Center, B0101-15-0552; DeepView] and Samsung Electronics Co., Ltd.

## Footnotes

[2]This is identical to DecoupledNet except that its classification and segmentation networks are trained with the same images, where image-level weak annotations are generated from pixel-wise segmentation annotations.

[3]We did not include the validation set for training and have less training examples than the competitors.

[4]http://cvlab.postech.ac.kr/research/decouplednet/

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
