[Reviews · NeurIPS 2015]

Submitted by Assigned_Reviewer_1

Nice work on semantic image segmentation with CNNs, focusing on the problem of model learning from a large number of image-level annotations and a small number of pixel-level annotations.

Key idea is to decompose the model into a classification convolutioanl network and a class-agnostic deconvolutional segmentation network. Back-propagation (up to the last pooling layer) in the classification network provides the per-class activations (saliency maps) that drive the segmentation network.

Another conceptually less exciting but practically rather important aspect of the proposed method is a data augmentation technique, which allows generating a large number of pixel-level segmentation maps from the limited set of training images.

My single key request to the authors in preparing the rebuttal/final version of the paper is to isolate the contribution of the data augmentation method. In particular, it is very important to know how much performance degrades when no such data augmentation is used, similarly to the previous methods in the literature that they compare with.

Overall the paper is well written. A minor request to the authors is to comment on the computational cost of their method. In particular, they should note that the back-propagation steps in Eq. (3) need to be performed multiple times (one time for each of the class labels present in the image), which adds to the computational cost of the method when a large number of object classes are present to the image.
Summary: Paper on learning CNN-based semantic image segmentation models from a large number of image-level annotations and a small number of pixel-level annotations. Key idea is to decompose the classification and segmentation stages. Significantly exceeds SOA on the challenging PASCAL VOC 2012 segmentation task when only very few strongly annotated images are used for training. Good solid work.

Submitted by Assigned_Reviewer_2

This paper shows impressive results on semi-supervised semantic segmentation by using a two-part network, where the first stage performs image classification, and the second part predicts a segmentation for each of the predicted labels. Bridging layers produce class-specific activation maps out of the classification network and these are used by the segmentation network.

The results are very good, since with even a few strongly annotated examples, the algorithm goes a long way. The approach is also quite novel, especially the idea of the bridging layers. Semi-supervised segmentation is an important enough problem for this result to be significant.

There were, however, a couple of details and ablation studies missing: 1) Twice in the paper (lines 266-267 and lines 310-311) the activation map g is produced from a set of labels instead of a single label. How does this work? Does one simply sum up the f^l_{cls} of all the labels one is interested in and feed that into the bridging layers? 2) The paper does two interesting things whose impact is not evaluated: first, it frames the segmentation as a binary labeling problem, instead of a multiclass problem which is what everyone else does, and second it uses a rather sophisticated data augmentation scheme (described in lines 307-315). It would be good to have ablations investigating how much each innovation buys.
Summary: This paper shows impressive results on semi-supervised semantic segmentation by using a two-part network, where the first stage performs image classification, and the second part predicts a segmentation for each of the predicted labels. The architecture is novel and the results are impressive. The experimental comparison to alternative approaches is convincing,

Submitted by Assigned_Reviewer_3

This paper addresses one of the most central problem of computer vision: semantic segmentation of natural images. The hardness of labeling ground-truth for this task makes purely supervised approaches very expensive. So it is of utmost importance to transfer the knowledge of whole image labeling models to segmentation.

The paper presents an extremely simple, very elegant and efficient approach to semantic segmentation in natural images: the classifier network is trained with a large amount of training data labeled with whole image annotations. The classification output is backpropagated to one of the intermediary layers and these class-specific backpropagated activation maps are used as input to training the deconvolution network for segmentation. Given its well motivated, but very simple nature this approach can reach very high quality results even with constrained amount of segmentation ground-truth.

The paper is clearly written and tackles one subclass of the very fundamental class of machine learning tasks, even beyond computer vision. The methods described in the paper can be useful in a lot of very different looking use-cases as a more efficient alternative to multi-instance learning, so in my opinion the potential impact of the approach is very high.
Summary: This is one of my favorite computer vision paper of the past year.

This paper gives a very simple but efficient approach for the

fundamental problem of semi-supervised segmentation.

The generic idea of the paper

has a huge potential of becoming the most crucial component for top performing segmentation and detection results in the coming years, and could also serve as a template for other weakly labeled tasks.

Submitted by Assigned_Reviewer_4

This paper describes a method of semantic segmentation, that can be applied particularly in cases where there are many image-level class labels but few segmented images.

Classes are first identified using a classification network, then individually segmented using a second segmentation network.

The second network is fed with the upper convolutional feature map of a the classification network, as well as the back-propagated activation of the individual class, which acts as a sort of seed.

These are passed through a set of fc layers, which are reshaped and passed to a generating ``deconvolutional'' network that produces the segmentation for the class.

Evaluations are performed using Pascal VOC.

Overall, this pieces together some good components to create a nice segmentation system.

However, some of the design choices could be further explored, and it would be nice to compare on more datasets than only Pascal VOC 2012.

Questions follow:

* I believe the same segmentation network is applied to all classes (i.e. weights are shared between classes), from l.219.

But how much does this fact contribute to its performance when using few strongly annotated images, compared to the combinatorial data augmentation?

* How many channels does the bridging layer output g_i^l have?

Fig 2 suggests just 1.

* While Fig 2 shows the bridging layer output is similar for each class, I wouldn't necessarily conclude that the "search space may not be as huge as our prejudice".

Another possibility is that the class pattern is used in the first layer of the deconvolutional network to essentially perform the same unpooling (with switches) operation that forms the backpropagated map f^l_cls, so that the supplying the class provides more information than the spatial component (which is largely redundant).

* Related, what happens if only f^l_cls is used (no f_spat), or if it is substituted with the class using a 1-hot encoding?

What is the contribution of using this particular representation?

* The text indicates the segmentation network is applied to each "identified label" -- how are these labels identified precisely from the classification net outputs?

* A few additional references for semantic segmentation with convnets, which might also be included:

- Farabet et al. "Learning Hierarchical Features for Scene Labeling" TPAMI 2013

- Eigen & Fergus "Predicting Depth, Surface Normals and Semantic Labels with a Common Multi-Scale Convolutional Architecture" arxiv 2014

- Wang et al. "Deep Joint Task Learning for Generic Object Extraction" NIPS 2014
Summary: Overall, this pieces together some good components to create an effective segmentation system.

However, some of the design choices could be further explored, and it would be nice to compare on more datasets than only Pascal VOC 2012.

Author Feedback
Author rebuttal: We appreciate insightful and constructive comments by reviewers.

1. [R6] Novelty
Most reviewers agree that the proposed decoupled deep network (DecoupledNet) for semi-supervised semantic segmentation is novel and interesting, and its performance is impressive. DecoupledNet stands out from the traditional DNN in that the network is composed of two parts for classification and segmentation and we train the two subnetworks separately using heterogeneous training data. We would like to make sure that our segmentation network is shared by all classes, not constructed for individual classes.

2. [R1, R2, R7] Control experiments
To isolate contribution of the proposed decoupled architecture, we additionally trained our network without proposed data augmentation technique--combinatorial cropping. Specifically, we construct binary ground-truth segmentation masks by setting one class in the image as foreground at a time and the rests as background. Then for each foreground class, we simply crop multiple sub-images containing the class to construct training data. Without our data augmentation, the performance is degraded by about 1.5% point in mean IoU for all three different semi-supervised cases. This result supports that our algorithm is much better than other comparable methods even without combinatorial augmentation while our data augmentation technique further improves performance.

3. [R1] Computational cost
It is true that computational cost of our network increases as more classes are associated with an input image. However, this additional cost is not significant in practice since inference for a single class takes about 60 ms and real-world images often contain only small number of semantic classes (3.5 classes per image in average even in MS-COCO dataset). Note that we need to compute f_{spat} only once per image.

4. [R2] Evaluation on other datasets
We evaluate the performance on PASCAL VOC dataset since it is the most commonly used benchmark in the recent literature; other datasets are outdated and it is difficult to find the results of comparable algorithms. We plan to publish our models and codes to facilitate other researchers to test our algorithm on various datasets.

5. [R2] Our conjecture about search space size of segmentation network
We do not mean that we train bridging layers to produce coherent outputs within each class. Rather, we describe the observation that g^l_{cls} tends to exhibit small intra-class variations and large inter-class variations probably because class-specific saliency map f^l_{cls} is provided as one of the inputs to bridging layer. We expect that this property is beneficial to train the segmentation network with limited number of strong annotations. We will clarify this if our paper is accepted.

6. [R2] Contribution of f_{spat} and f^l_{cls}
The class-specific saliency map f^l_{cls} tends to highlight discriminative activation patterns in each class, which may be insufficient for accurate segmentation since activations from non-discriminative regions may also be important for shape generation. On the other hand, outputs from classification network f_{spat} typically captures abstract information of an input image useful for shape generation. We believe that the combination of the two kinds of information would be useful to obtain accurate segmentation map. In addition, we believe that f^l_{cls} encodes richer class-specific information than one-hot encoding, since spatial configuration of the activations preserved in f^l_{cls} provide useful cues for segmentation.

7. [R2] Number of channels in g^l_i
Thanks for the remark. The number of channels for g^l_i is same as that of f_{spat} and f^l_{cls}, which is set to 512 in the proposed architecture. Figure 2 visualizes a selected channel of g^l_i, and the visualizations of other channels are found in our supplementary material. (We believe that our webpage style supplementary material is very useful to understand the characteristics of our network.) We will clarify this issue if our paper is accepted.

8. [R2] Identifying relevant class labels with classification network
If the output probability of a class label is larger than the predefined threshold (0.5), the label is regarded to be relevant to the input image.

9. [R2] Additional references
We will add the discussion about the suggested papers if our paper is accepted. Thank you.

10. [R7] Constructing activation map g for multiple labels
In principle, activation map g for multiple labels can be obtained by summing up f^l_{cls} for all classes and feeding it to bridging layers. Alternatively, we can efficiently obtain the same activation map through back-propagation with all identified labels set to 1 and the others set to 0. We chose the second option due to memory efficiency, but believe the first option is also fine.

11. [R3] Presentation
We will clarity data augmentation procedure if our paper is accepted.